# A validated lineage-derived somatic truth data set enables benchmarking in cancer genome analysis

Megan Shand [1✉], Jose Soto[1], Lee Lichtenstein[1], David Benjamin[1], Yossi Farjoun [1], Yehuda Brody[1,2], Yosef Maruvka[1,3], Paul C. Blainey [1,4,5] & Eric Banks[1]

Existing cancer benchmark data sets for human sequencing data use germline variants, synthetic methods, or expensive validations, none of which are satisfactory for providing a large collection of true somatic variation across a whole genome. Here we propose a data set, Lineage derived Somatic Truth (LinST), of short somatic mutations in the HT115 colon cancer cell-line, that are validated using a known cell lineage that includes thousands of mutations and a high confidence region covering 2.7 gigabases per sample.

[1] Broad Institute of Harvard and MIT, Cambridge, MA, USA. [2] Klarman Cell Observatory, Broad Institute of MIT and Harvard, Cambridge, MA, USA. [3] MGH Cancer Center and Department of Pathology, Boston, MA, USA. [4] MIT Department of Biological Engineering, Cambridge, MA, USA. [5] Koch Institute for Integrative Cancer Research at MIT, Cambridge, MA, USA. ✉email: mshand@broadinstitute.org

Detecting somatic variation from whole-genome sequencing is essential to the understanding and treatment of cancer[1–3]. However, discovering somatic short variants (SNVs and Indels) with high precision and sensitivity is still a challenge owing to tumor heterogeneity, sequencing artifacts, mapping artifacts, and contamination from normal cells[4–6]. Many data pipelines and variant calling methods still disagree in a large number of sites, making it unclear which discrepancies are true variants and which are false[7]. Even variants that are called by multiple methods are not guaranteed to be true positives. This demonstrates a critical need for high-quality benchmarking data that could be used to disambiguate the discrepancies.

Many benchmarking data sets, including ICGC-TCGA DREAM, simulate variation by modifying bases in sequenced reads to known alternate alleles at various allele fractions[8–11]. Even with sophisticated modeling, these simulated mutations do not follow the true biological and physical pathways that generate real somatic mutations. As such, synthetic truth data penalize callers that model somatic variation better than the simulations. Other benchmarking data sets often combine germline samples (in silico or in vitro) to simulate heterogeneous tumors. However, germline variation is inherently different than somatic variation in nucleotide substitution frequencies, context, and genomic location frequency[1]. In addition, if germline variants are used as truth data, somatic variant calling pipelines must disable normal germline filtering. Benchmarking methods that use actual somatic mutations involve expensive validations of individual sites[8]. This is limited to a small number of sites and therefore is not powered to make good, unbiased estimates of the performance[4]. Another technique of using deeper sequencing[12] as validation, whereas more sensitive to low allele fraction sites, is no less prone to errors from sequencing, library preparation, or mapping artifacts.

Here, we provide a benchmarking data set of validated somatic mutations, Lineage-derived Somatic Truth (LinST), in a human colon cancer cell line with a DNA polymerase epsilon (POLE) proofreading deficiency (HT115)[13]. A known lineage tree structure that was determined using lineage sequencing (LinSeq)[13] is used to validate somatic variation across 11 whole genome HT115 samples, and to construct the high confidence region. LinST will benefit developers and consumers of somatic mutation calling software as the first truth set of its size consisting of real somatic mutations. These truth data are generalizable beyond understanding the mutations in the single HT115 sample in the same way that the characterization of NA12878 by Platinum Genomes[14] is used as truth data for training and testing new algorithms and models for germline variant calling for all samples. Although LinST is characteristic of colon cancer (as shown by the mutational signature), it still contains a broad spectrum of mutations which is helpful for assessing an algorithm that calls somatic mutations for any cancer type. Beyond the true mutations included in LinST, there are gigabases of region we are confident do not have somatic mutations (including regions enriched for artifacts). Providing this resource as a test of false-positive rates is important for all cancer types. In addition, the method for producing LinST could be used on any cell line that will accumulate mutations across individual divisions and could therefore be used to generate a breadth of truth data across many cancer types in the future.

## Results

LinSeq uses imaging technology to record a lineage structure by observing a single cell as it divides over multiple generations. Each of the nodes in the tree (Fig. 1) represents a single cell dividing into two separate sublineages. Once a sufficient number of generations have been observed, several cells are grown up separately and Whole Genome Sequenced to 35× coverage. Variant calling pipelines (see Methods section below) are then run on each of these "leaf" samples and compared with each other to validate the high confidence "branch" variant calls and concurrently define the high confidence regions.

Because the different samples contain partially overlapping variants (owing to the inheritance structure), the truth set (and high confidence region) depends on the particular tumor-normal pair of samples that are going to be benchmarked against it. Creating the truth set would therefore make use of the inheritance tree and also the calls that were made against all the bulk-sequenced samples.

To create a tumor-normal pair at a desired purity, we take three sequenced samples: a "pure tumor" sample and two sister samples that are distant to the tumor. The two sister samples are considered "normal" (relative to the "pure tumor"). We informatically mix one of the normal samples with the "pure tumor" to create the case tumor sample, and use the other normal as the case normal, for somatic variant discovery pipelines that are run with the matched normal (Fig. 1). Two sister samples are needed

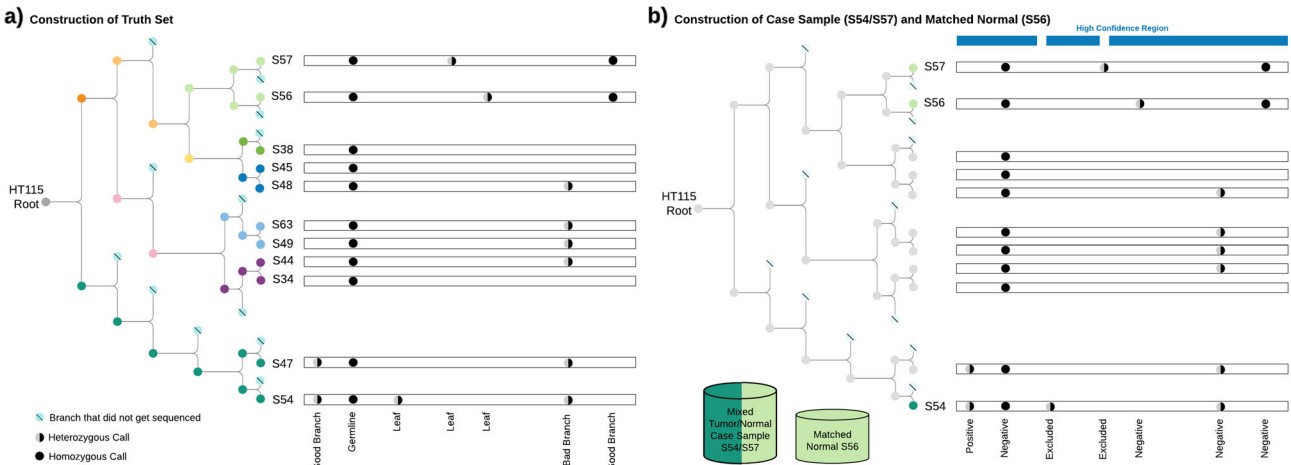

**Fig. 1 Lineage tree structure for HT115 and an example genomic region. a** Although most of the Good Branch Variants are heterozygous owing to single cell bottlenecks, we do observe some Good Branch Variants that are homozygous variants due to loss of heterozygosity events that occurred after the mutation arose or large scale deletions. **b** A case sample and matched normal are created from mixing any two samples together. A matched normal is taken from the closest relative to the mixed in normal. This results in a list of positive sites and excluded regions.

**Table 1 Counts of verified events by type.**

| Verified mutations | S57/S56 | S38 | S45/S48 | S63/S49 | S44/S34 | S47/S54 | Unique |
|---|---|---|---|---|---|---|---|
| SNV | | | | | | | |
| Coding | 50 | 32 | 32 | 70 | 52 | 45 | 24 |
| Non-coding | 1738 | 1656 | 2124 | 2825 | 2708 | 3267 | 1265 |
| Intergenic | 1793 | 1764 | 2574 | 2928 | 2847 | 3717 | 1398 |
| INDEL | | | | | | | |
| Coding | 0 | 0 | 0 | 0 | 0 | 0 | 0 |
| Non-coding | 255 | 218 | 308 | 264 | 255 | 387 | 154 |
| Intergenic | 193 | 158 | 293 | 287 | 305 | 441 | 161 |
| CNV | | | | | | | |
| Genic | 0 | 0 | 0 | 1 | 0 | 2 | 3 |
| Intergenic | 2 | 1 | 1 | 1 | 2 | 0 | 3 |
| Mulitple | 0 | 0 | 0 | 0 | 0 | 2 | 2 |

The unique CNVs verified by the lineage tree structure cover 27 megabases with one region at copy ratio 2, one homozygous deletion, and the other six at copy ratio 1.5. "Multiple" type CNVs refers to an event that spans multiple genes and intergenic region. "Non-coding" refers to intronic or other non-coding regions within a gene.

to act as the normal sample because there is not enough coverage in one sample to use as both a mixed-in normal in addition to a matched normal. With deeper sequencing, only one sample would be needed to act as normal tissue.

To generate the truth set for each tumor-normal pair of samples, SNVs and Indels are filtered using the lineage structure. We define a Germline Variant as a site that is called in all 11 samples. This is either a germline variant of HT115 or a somatic mutation that occurred before the LinSeq process began. For the purposes of the truth set, these sites are considered germline because they occur in both the "tumor" and "normal" sample, and therefore the site is not included in the truth set but remains in the high confidence region. If a somatic variant discovery pipeline calls these variants, they are considered false positives.

We define a Leaf Variant as a site that is called in only one of the 11 samples. These variants could be real somatic mutations that occurred after the single leaf cell was grown for bulk sequencing, or they could be artifacts that only occurred in one sample by chance. These variants can occur at arbitrary allele fractions because the bulk sequencing process requires growing many cells and can therefore have various subclones within the cell population. Owing to these uncertainties, Leaf Variants are removed from the high confidence region. If a variant discovery pipeline makes calls at these sites, they do not count as true positives or false positives.

We define a Branch Variant as a site that is called in more than one but fewer than 11 samples. A Good Branch Variant is defined as a Branch Variant for which all samples with this variant share a single common ancestor, and none of the other samples share that ancestor. A Good Branch Variant that is called in the tumor sample but not the normal is included in the truth set. If a somatic variant discovery pipeline calls these variants, they are considered true positives. These are defined as lineage structure concordant branch variants in the original LinSeq paper[13]. A Bad Branch Variant is defined as Branch Variants that are not Good Branch Variants. These sites are not included in the truth set, but are not excluded from the high confidence region. If a somatic variant discovery pipeline calls these variants, they are considered false positives.

By only removing Leaf Variants called in the tumor or the mixed-in normal from the high confidence region, we are able to retain sequencing and mapping artifacts in the high confidence region of the truth set (Bad Branch Variants that we are confident are not real somatic mutations).

The true positives generated from this technique are real somatic mutations, so pipelines can and should be run exactly as they would be on real samples (with germline and matched normal filtering). These cell-lines also have copy number events that have been validated with the tree structure or are seen in all leaf samples. LinSeq can also be used to validate these CNVs (although at a smaller number of sites than SNVs and Indels, see Table 1). Using these CNV calls, we adjust our filtering strategy for SNVs and Indels to account for the expected number of copies in each amplified region.

To demonstrate the consistency of LinST, we made all possible pairwise mixtures at three simulated purities, 10%, 20%, and 50%, and ran two somatic variant discovery pipelines (Mutect2[15] and VarScan2[16]) on all mixtures. Across all of these mixed samples, the sensitivity and false-positive rate metrics (measured using LinST) are consistent given sequencing depth and purity (Fig. 2). This demonstrates that LinST provides reasonable and consistent measurements of true-positive and false-positive rates for somatic discovery methods. By keeping Bad Branch Variants within the high confidence region, we see an average 2% increase in the number of false positives for each mixture. Keeping as many difficult genomic regions within the truth set as possible gets a more-accurate estimation of the real false-positive rate.

The Good Branch Variants match previously published bulk POLE mutant colon tumor sample data[13] and are consistent across each leaf sample (Fig. 3). Compared with a synthesized truth data set, ICGC-TCGA DREAM[8], the mutational signature of the Good Branch Variants from LinST is much more realistic to a tumor sample (Fig. 4). The mutational signature of each leaf sample aligns with signatures observed in COSMIC, SBS10a and SBS28 (particularly from the high levels of C/A and T/G mutations, respectively). This is expected as SBS10a is associated with POLE mutations and SBS28 commonly occurs in colorectal cancers with POLE mutations. SBS28 is frequently observed together with SBS10a[17,18]. In addition, LinST true mutations have a lower proportion of intergenic sites than ICGC-TCGA DREAM. When normalized by the number of reference bases in each classification category, LinST has a more uniform distribution of mutations across categories than ICGC-TCGA DREAM.

## Discussion

LinSeq could be repeated on other cancer types and samples to generate other benchmarking data sets, with the potential for testing various wet laboratory techniques as well. This could be achieved by sequencing each leaf sample twice—once with a control wet laboratory technique and once with an experimental wet laboratory technique. Choosing a tumor sample for the root with a matched normal would allow us to disentangle germline variation from somatic mutations that occurred before the observation of the lineage tree. Bulk sequencing the HT115 sample from which the

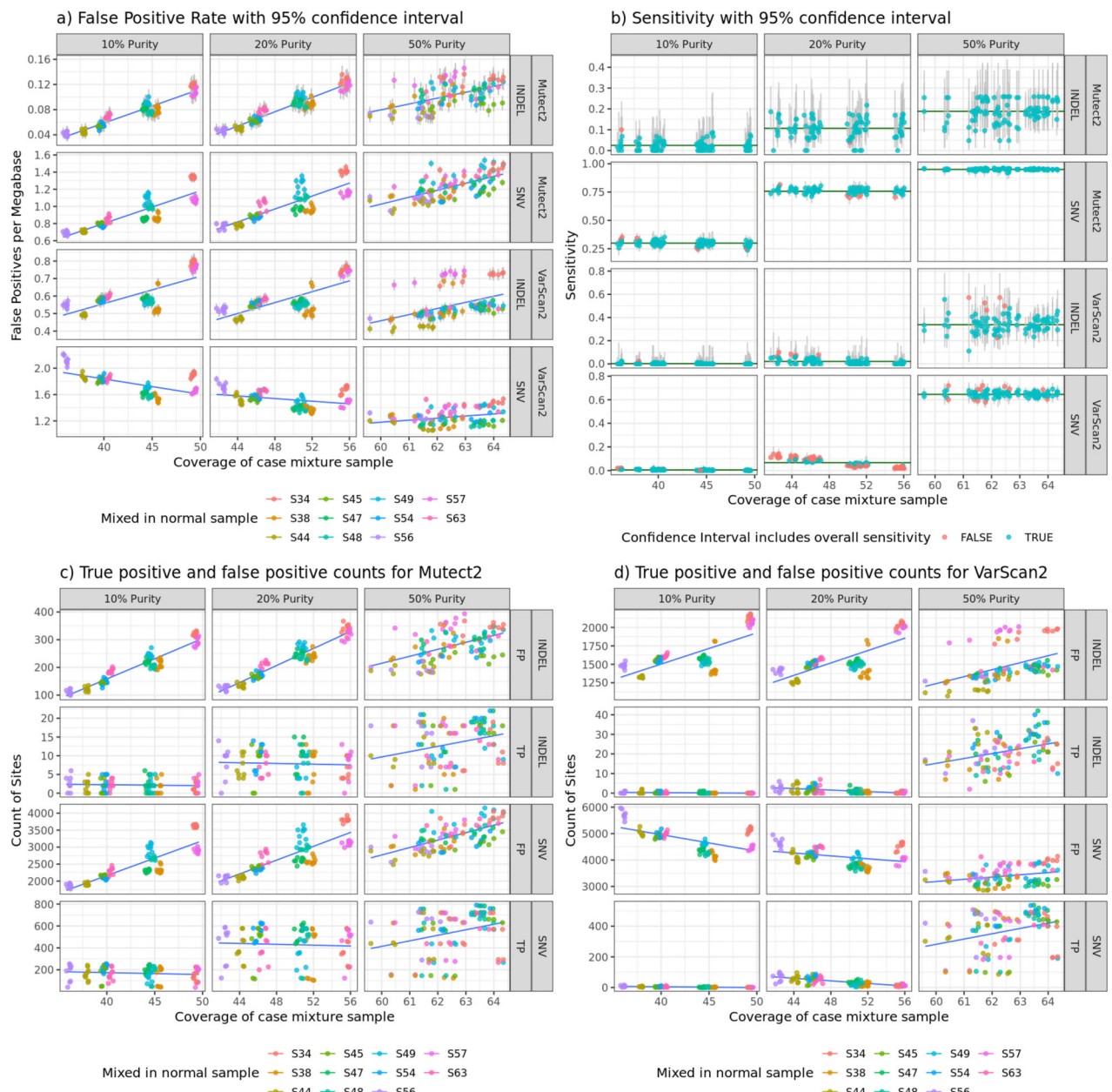

**Fig. 2 False-positive rate, sensitivity, and counts of true-positive and false-positive calls from all possible mixtures run with Mutect2 and VarScan2.**
**a** False-positive rate clusters by the mixed-in normal sample at lower purities. This is owing to variability across samples in quality and false-positive rate. **b** Sensitivity mostly follows the expected model for each purity. Green horizontal line denotes overall sensitivity calculated across all mixtures. **c** Count of true positives and false positives called by Mutect2. **d** Count of true positives and false positives called by VarScan2.

root cell was taken could also be used as a matched or mixed-in normal. Although this was not performed in the original LinSeq experiment, a cleaner version of LinST could be generated using this sequencing method. This would allow the somatic variant discovery pipelines to be run with the real matched normal or a bulk sample taken before the root rather than using a separate leaf sample, thus removing a complication. Even with the data presented here, more mixtures could be made between three or more samples to simulate tumor heterogeneity.

Deeper sequencing of the samples we have would provide the benefit of obtaining even lower purity mixtures, along with more realistic high coverages. It would also give us enough reads to be able to use the same sample as the mixed-in normal and the matched normal, removing another complication in the pipeline and making the purity more realistic. Cross-platform sequencing

technologies could be used on the leaf samples, in addition to Illumina sequencing, to further validate the Good Branch Variants while mitigating Illumina specific sequencing artifacts.

To replicate this experiment as a truly public data set, not only as sequencing data but as samples for sequencing, each leaf sample could be continued as cell-lines that could be shared. This would enable further testing of pipelines as well as wet lab techniques. However, the difficulty in this approach would be that somatic mutations would continue to accumulate in these samples over time, meaning that the high confidence region of the truth set would shrink. This might not have a large effect on the final benchmarking data set but should be considered regardless.

We hope that LinST and the methods to produce it will be useful for benchmarking new short somatic variant calling pipelines and generating further truth data for the general community.

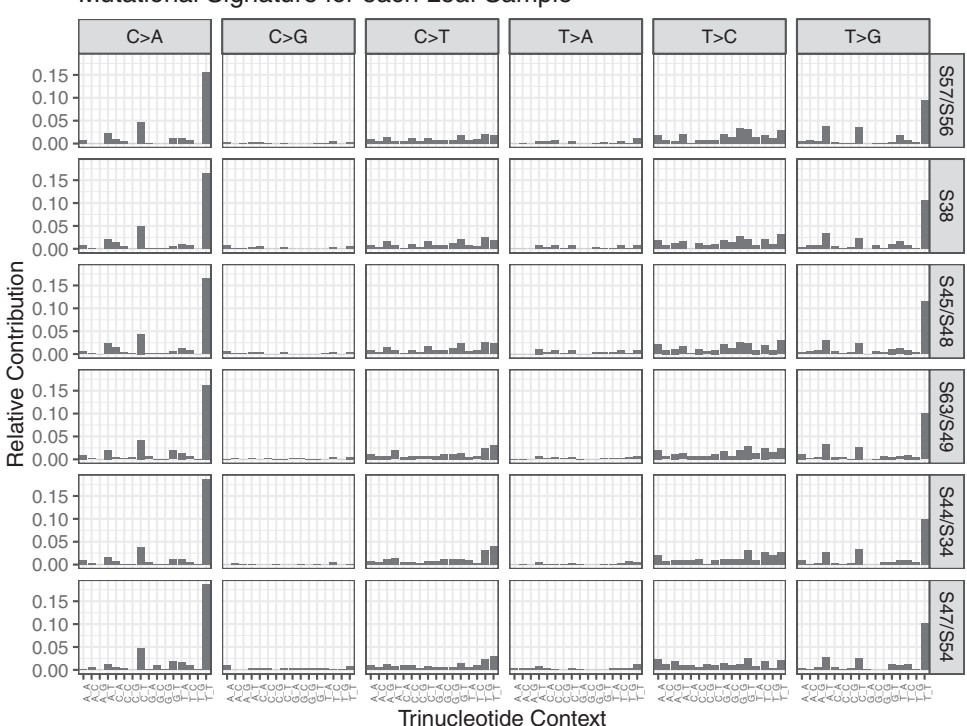

**Fig. 3 Mutational signature across each leaf sample is consistent.** Samples are shown with their sister sample since the lineage tree is used to validate the mutations there are no differences in validated sites between sister samples.

## Methods

**Primary analysis.** All data were obtained from the original LinSeq experiment by Brody et al.[13]. Data can be downloaded from https://www.ncbi.nlm.nih.gov/sra/SRP159787. HT115 was purchased from the Broad Cancer Cell Line Encyclopedia. Cell identity was authenticated by the Broad Genomics Platform for HT115 using previously stored fingerprint genotypes. The fingerprint consists of the genotype at 82 loci from the query sample, which were compared against fingerprints of all the Cancer Cell Line Encyclopedia cell lines using the farthest neighbor graph algorithm. The highest correlation (0.83) was found between our HT115 sample and the CCLE HT115 sample. There was no Mycoplasma contamination observed in the sequence data.

Samples were reverted and realigned to GRCh38 human reference using the GATK best practice recommendations[19]. Reads were aligned with BWA-MEM[20], and duplicate reads were marked using Picard. Quality scores were then recalibrated using GATK base quality score recalibrator. Data were stored in per-sample BAM format. The mixtures were generated by combining samples that were individually downsampled using SAMtools[21].

**CNV calling pipeline.** GATK's Model Segments pipeline[19] was used to generate somatic CNV calls. Each sample was run individually with a panel of normals generated from 60 whole-genome normal samples from The Cancer Genome Atlas (TCGA) sequenced at the Broad Institute Genomics Platform. The GATK version 4.1.2.0 pipeline was used with the penalty factor set to 5 to combat over-segmentation owing to low coverage (samples were sequenced to 35× coverage).

Copy number events were analyzed across all samples to determine whether they follow the tree structure. We obtained copy number amplifications and deletions that followed the tree structure and were called in either every sample or in one leaf sample. While we found CNVs that do not follow the tree structure, it is unclear whether these are due to false-positive copy number calls in some samples or false negatives in other samples.

These copy number calls that did not follow the tree structure were removed from the high confidence region if the tumor sample was not copy neutral in those regions. Copy number calls that only occur in the tumor sample were also blacklisted, as their quality is ambiguous. If the tumor sample is copy neutral, we maintained those sites in the high confidence region, as we still observe real SNVs that follow the tree structure that are unaffected by potential copy number events in other samples (this was confirmed by manual inspection).

We observed eight unique CNVs that occur within the known lineage tree structure and 646 unique CNVs that occur across all 11 leaf samples. We also observed copy neutral loss of heterozygosity in all samples in some regions. This did not affect our short variant analysis as we could still expect somatic mutations to occur in these regions after the loss of heterozygosity, at 50% allele fractions. We

do not suspect any genome doubling events in these cells from manual review of the CNV calls.

**SNV and indel calling pipeline.** There are two steps of SNV and indel calling pipelines—1) detecting germline or Good Branch Variants, and 2) detecting Leaf Variants. First, the pipeline for detecting potential true variants is designed to be precise. True variants are expected to follow the diploid assumption (in regions without copy number events) because any mutation went through a single cell bottleneck in the tree structure, and therefore its allele balance should be consistent with being either a hetrozygous or homozygous variant. We only consider variants with good allele balance as eligible candidates for being true variants. Second, the pipeline for detecting mutations that occurred after the observation of the tree structure (Leaf Variants) is designed to be sensitive even to variants that do not follow the diploid assumption. Each leaf sample was grown up by the laboratory in order to achieve whole-genome bulk sequencing. For that reason, there could be subclonal populations in each leaf sample, whereas the Good Branch variants will be monoclonal. Ambiguity about the source of Leaf Variants is the rationale for excluding these sites from the high confidence region. We use HaplotypeCaller[19] with stringent filters to detect the potential true variants and Strelka2 in somatic mode[22] to detect the potential Leaf Variants. The choice to use Strelka2 is somewhat arbitrary, any sensitive somatic variant calling tool could be used to detect potential Leaf Variants. Here we chose a pipeline that we did not use to measure sensitivity and precision with the final LinST data set (in this case Mutect2[15] and VarScan2[16]) to reduce the chance of artificially increasing precision by testing the same pipeline that is used to generate the truth data.

HaplotypeCaller was run in joint-calling mode, meaning all reads shared local assembly, with all 11 leaf samples as input in order to increase sensitivity. Standard GATK best practices[19] using Variant Quality Score Recalibration was run as an initial filtering strategy. We then filter out any site where any sample has a Genotype Quality <25. Although the majority of sites are expected to have heterozygous genotypes owing to single cell bottlenecks, we did not filter out non-heterozygous sites explicitly. Regions with deletions still have homozygous variant genotypes and regions with amplifications were filtered based on the number of copies. Given the copy number call at any site (taken from the CNV calling pipeline described above), any heterozygous site whose Allele Depth is <1% percentile from $Binomial\left(n = \text{depth}, p = \frac{1}{\text{ploidy}}\right)$ in more than half of the samples is filtered out. This filter is not applied to homozygous variant genotypes, however all other filters are applied. Any site that is "no-called" for any sample is also filtered out. Finally, all of these passing variants that do not follow the expected lineage tree are removed from the list of true variants (but remain in the high confidence region). The known tree structure gives the ability to validate each of the sites across multiple samples separately from individual filters based on read signatures.

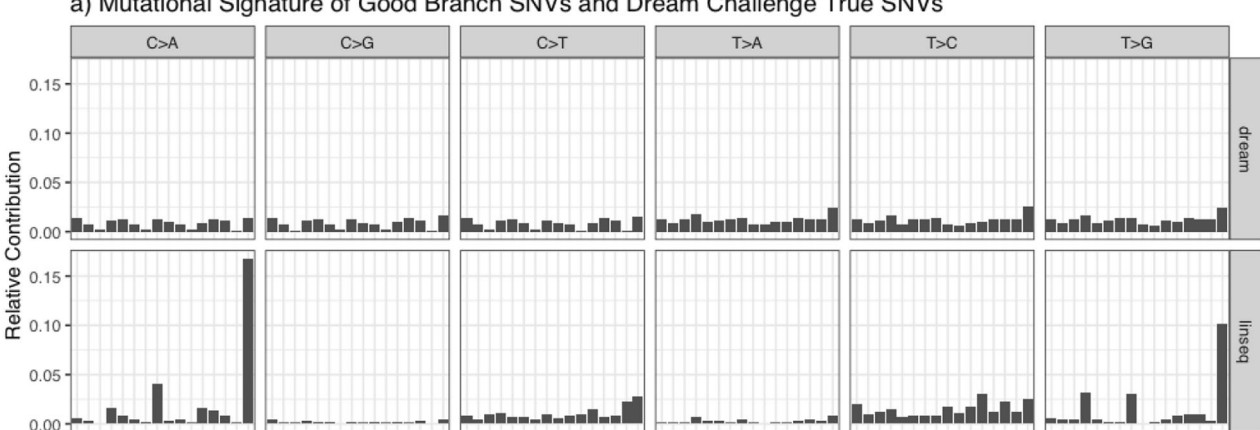

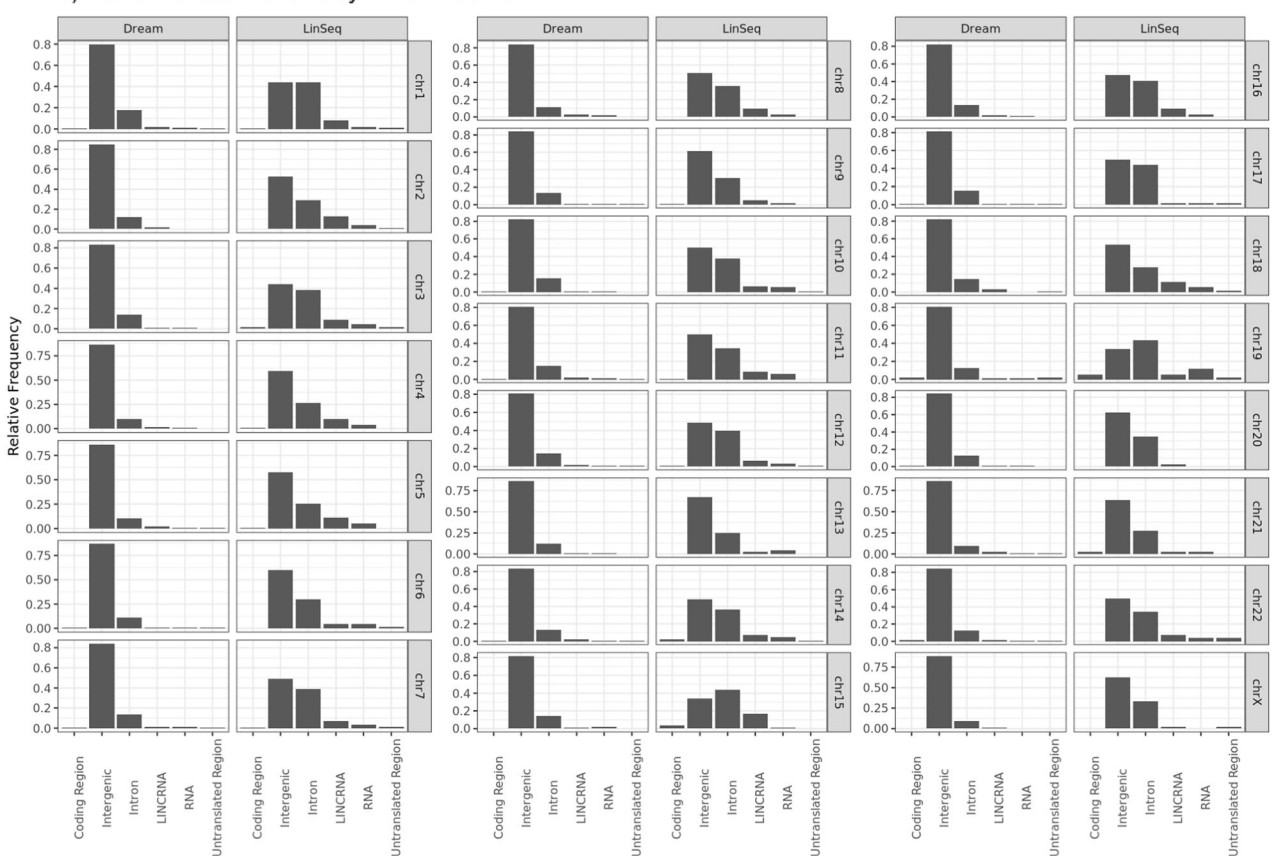

**Fig. 4 Mutational signature and variant classification by chromosome of Good Branch Variants compared with ICGC-TCGA DREAM true sites.**
**a** Mutational Signature of the Good Branch Variants shows a distinct signature with high levels of *C/A* and *T/G* mutations. ICGC-TCGA DREAM true sites are uniform across various contexts. **b** Variant classifications by chromosome, normalized by the number of bases in the reference in each category, show higher than expected proportion of intergenic sites in ICGC-TCGA DREAM than LinST. If each base in the genome was equally likely to be a true site in this plot, all bars would be the same height.

Genotype information is not taken into account when comparing alleles across the tree structure, so heterozygous and homozygous variant genotypes are all included. The list of passing variants at this point are considered the true somatic variants for each sample (Table 1).

The Strelka2 somatic pipeline is run with each possible pair of leaf samples, one acting as the tumor and the other acting as the matched normal. Thus each of the 11 leaf samples is run through the pipeline as a tumor 10 times, once with each other leaf sample acting as the matched normal, generating up to 10 calls at each site, some of which may be filtered (by Strelka2). If more than half of the calls at

any site pass Strelka2's filters, then the site is included in the potential leaf calls. This is in order to balance the risks of incorrectly calling an artifact a Leaf Variant. If half or more of the calls for one sample at any given site are passing, overall the Strelka2 pipeline believes that call, so we include it as a potential Leaf Variant. The one exception is if all 10 runs are called (filtered or not), the site is included in the potential leaf calls. This is because for many sites called by Strelka2 in all 10 runs, even though they looked like sites with borderline evidence in each "tumor-normal" pair, overall they looked like they could be real somatic variation. Finally, when looking at these consolidated filtered calls across all 11 leaf samples, if the call

is only found in one sample it is kept as a Leaf Variant and removed from the high confidence region.

**Comparison of SNVs and indels.** In order to assess the quality of LinST, the Mutect2[15] and VarScan2[16] pipelines were run on each possible pairing of samples at various mixture rates (to simulate various purities). We then checked that sensitivity and FPR for these mixtures run with each variant discovery pipeline are consistent and reasonable. The pipelines were run with a matched normal taken as the sister sample to the mixed-in "normal" sample. This ensures that a similar sample is used as the matched normal to the diluting normal sample, but not the same reads. The VarScan2 pipeline was run with default parameters first running the somatic tool for variant discovery, followed by the processSomatic tool to filter to high-quality sites. The Mutect2 pipeline was run with defaults for variant discovery and filtering as well as with a panel of normals made from 1000 Genomes Project samples[23] (https://storage.googleapis.com/gatk-best-practices/somatic-hg38/1000g_pon.hg38.vcf.gz), and gnomAD common germline variant filtering[24].

The resulting VCFs were compared within the high confidence region to the truth set using rtgtools VariantEval[25] for both SNVs and indels to determine each pipeline's sensitivity and FPR for each mixture.

We take all possible mixtures that will provide true sites at simulated purities of 10%, 20%, and 50%. Simulated tumor purity does not mean a single expected allele fraction of all sites in a given mixture. We expect the majority of sites to have an allele fraction that is half of the simulated purity, but due to CNVs and copy neutral loss of heterozygosity events, some sites will be at much lower allele fraction, and others will be at the purity fraction itself. We have $\binom{11}{2} - 5 = 50$ possible pairings as we are picking pairs of samples from the 11 sequenced samples, but we cannot use the five pairings of sister samples since there are no variants different between them that we can use the lineage tree to validate ([S57/S56], [S45/S48], [S63/S49], [S44/S34], and [S47/S54]). In addition because S38 has no sister sample, there are no Good Branch mutations that are not also present in the [S45/S48] branch, which removes another pairing (50−1 = 49). Within each pairing there are two possible mixtures (sample 1 as the tumor and sample 2 as the normal or vice versa). This results in $49 \times 2 = 98$ pairings, each at three purities or $98 \times 3 = 294$ mixtures.

In all, 95% confidence intervals were derived for each mixture's sensitivity using the exact Clopper–Pearson method, as the number of true positives and false negatives is small enough to use an exact method. 95% confidence intervals for FPR use the asymptotic normal approximation for the binomial distribution, as the larger number of false positives and true negative megabases meets the assumptions for using an asymptotic approximation (Fig. 2).

Mutational signatures of the Good Branch Variants and ICGC-TCGA DREAM were calculated with MuSiCa[26]. Variant classification was annotated using GATK's Funcotator pipeline[19] (Fig. 4). Classifications were derived from GENCODE 28 reference annotations[27].

**Number of sites per branch.** Another way to demonstrate the validity of using the LinSeq lineage tree is to blind ourselves to the tree structure and assess the variants before using the tree as a filter. After applying all filters except the lineage tree we have a list of high-quality variants. We can compare the number of these high-quality variants that are Good Branch Variants to those that are Bad Branch Variants as another way to check the validity of the lineage tree itself. After removing chromosome 4 owing to a large loss of heterozygosity event that occurred in the [S49/S63] branch, and all sites with allele frequency >1% in gnomAD v2.1[24] to remove common germline variation, a Wilcoxon rank-sum test with continuity correction was performed by ranking each subset of samples based on the number of variants called within that subset (two-sided $p$ value = 9.0e-8). The alternative hypothesis is that the true location shift between the ranks of those subsets that are consistent with the tree and the ranks of those subsets that are inconsistent is not equal to zero. The test statistic is $W = 1270$, sample size for consistent sets is $n = 9$, sample size for inconsistent sets (with at least one variant call) is $n = 142$, estimated difference in location is 301 with a 95% confidence interval of [170, 316]. (Fig. 5). Those subsets that are inconsistent with the tree with the most variants have a subset size of 10 samples and are owing to Germline Variants being missed in a single sample (note that the number of Germline Variants is around 5 million). Both Germline Variants called in all 11 samples and Bad Branch Variants, are excluded from the truth set but remain in the high confidence region (as neither artifacts nor true germline variation are true somatic variation).

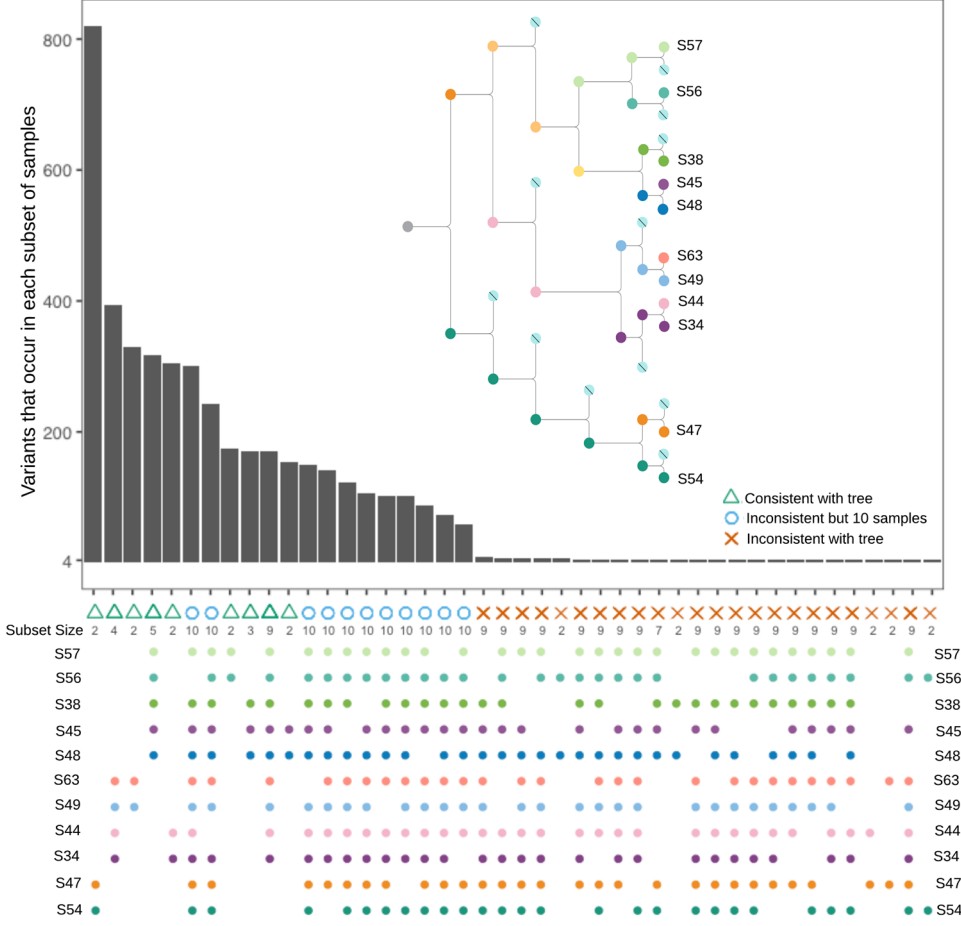

**Fig. 5 Number of variants called by HaplotypeCaller within each subset.** There is a clear separation of subsets that are consistent with the tree structure.

**Alternative lineage methods**. Another similar approach to LinSeq to determine a cell lineage could be to introduce bottlenecks in the cell lineage without using the imaging technology LinSeq uses to determine the lineage tree, but use other methods to discover the lineage tree[28]. This might be a simpler approach in the wet laboratory, but warrants further investigation if used to generate benchmarking data sets. The hierarchical clustering using distances based on allele fraction in mitochondrial variants from Ludwig et al.[29] was not able to rediscover the full known lineage tree on this data set. Although this technique was designed for single cell sequencing, here we used the bulk sequencing from each leaf sample. From manual review we see that the signal from the low allele fraction mitochondrial variants was overwhelmed by noise. This is because the allele fraction of discovered sites is low enough that calls in some of the 11 samples are low quality and could be false positives or false negatives due to being filtered out incorrectly.

**Reporting summary**. Further information on research design is available in the Nature Research Reporting Summary linked to this article.

## Data availability
Illumina reads aligned to hg19[GRCh37] are available in SRA under accession number SRP159787. Each of the mixed bams, along with their matched normal bam is available for public use along with the matching VCF of true-positive sites and interval list of high confidence regions (a free account is required: https://app.terra.bio/#workspaces/broad-dsp-spec-ops-fc/somatic_truth_data_from_cell_lineage).

## Code availability
Code to generate truth data sets and mixtures, run VarScan2, and benchmark results is available at https://doi.org/10.5281/zenodo.4289043. Code for CNV pipeline is available at https://github.com/gatk-workflows/gatk4-somatic-cnvs/tree/1.3.0, the Mutect2 pipeline at https://github.com/gatk-workflows/gatk4-somatic-snvs-indels/tree/2.5.0, and the Mitochondria pipeline at https://github.com/gatk-workflows/gatk4-mitochondria-pipeline/tree/1.1.0.

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

## Acknowledgements
We are grateful to Chip Stewart, Takuto Sato, Christopher Kachulis, Mark Fleharty, Madeleine Duran, Sarah Walker, Andrea Haessly, Sam Friedman, and Kristian Cibulskis for helpful suggestions and feedback on this project.

## Author contributions
Y.M. conceived of this study; Y.B. and P.C.B. conceived and ran LinSeq and shared the data; J.S. and M.S. analyzed the data and developed the pipelines; M.S. drafted the manuscript; L.L., D.B., Y.F., E.B., and P.C.B. supervised the project. All authors helped to revise the manuscript.

## Competing interests
The authors declare the following competing interests: the Broad Institute and MIT may seek to commercialize the LinSeq technology, and related applications for intellectual property have been filed. P.C.B. is a consultant to and equity holder in companies in the microfluidics and life sciences industries including 10X Genomics, GALT, Celsius Therapeutics, Next Generation Diagnostics, Cache DNA, and Concerto Biosciences. The remaining authors declare no competing interests.
