## [Peer Review File · Communications Biology]

Reviewers' comments:

Reviewer #1 (Remarks to the Author):

Shand et al present a nice resource for somatic mutation variant calling (does the resource have a name? This might be useful). The goal is to overcome many of the limitations of previous synthetic resources for benchmarking somatic mutation calling, done by either mixing germline sequencing data or computationally spiking-in mutations. The main dataset is a known lineage of HT115 cells from the original LinSeq paper and their associated whole-genome sequencing data. The authors turn this into a resource for somatic mutation calling by mixing sequencing from cells (leaves of the tree) together to more accurately simulate heterogeneous tumor data. One challenging aspect of the paper is that it's hard to read without a lot of context from the original LinSeq paper. I have a couple of suggestions below to help make this more of a stand-alone effort, though there are probably many ways to do that. I also couldn't get the Terra link to load, so I'm not sure what sequencing datasets are available, but it would be nice if there were a full complement of calls for each leaf, all the mixtures, etc. to really highlight the resource aspect of this work. Overall this is a valuable resource for the community, but the paper needs significantly more details for end-users to find it useful.

Comments:

- How uniform are the clonal subpopulations? Given the cell line of choice here is a hypermutator, it would be nice to see summary stats on each leaf, mutational signatures (I would suggest anything but a 3d lego plot), and chromosome distributions. This is meant as a mutational calling resource, beyond the original LinSeq paper, and these are important details. Either a plot or an expansion of table 1. This would really help interpret figure 2.
- It's a bit of a can of worms, but could you expand on the "The allele fraction from Ludwig et al was not able to rediscover the known lineage tree on this dataset.." comment on line 245? It would be nice to understand how their approach failed here. Also if you're mentioning lineage it would probably be worth putting a review citation in for the emerging set of experimental lineage approaches.
- The crux of the paper is the paragraph on line 93, which really needs to be expanded: For instance line (95) "For each depth and purity, this results in consistent sensitives and false positives per megabase, see Fig. 2." is all that's mentioned. How does the FP rate compare to TP? Or the number of calls overall? Also I don't think 'sensitives' is the right word (or a word)?
- Can you compare Mutect's calling statistics (FP,TP etc) to another synthetic calling resource (maybe the dream challenge)? It would be good to demonstrate that this callset overcomes many of the issues in previous datasets, even if it's only in one aspect.

Reviewer #2 (Remarks to the Author):

In this study, Shand et al. generated a somatic truth set of somatic alterations from previously generated data from a colon cancer cell line. The development of strong somatic truth sets for analysis of WGS data continues to be an area of need. The use of more comprehensive analyses will be necessary and thus involves the use of cross-platform data to address and minimize deficiencies associated with sequencing platforms. However, this study does not address this aspect. The use of LinSeq data is interesting but is not clear how useful this data set would be for the research community, unless the data is meant to be a reference that can be used specifically for basic research studies utilizing the HT115 cell line.

Major comments:

The use of tumor/normal pairs is a bit unclear and perhaps a revision to Fig1 would be helpful. Given that the goal of the study is to produce a somatic truth set, a major concern is that a true paired normal to HT115 was not sequenced/analyzed. Please clarify. Given cross-generational genetic variation, capturing these changes in tumor and normal cells will be important for defining a somatic truth set. Along these lines, the reasoning for using distant sister tumor samples as "normals" is unclear.

If the truth set is aimed to be used for colon cancer studies, comparison of the data against reported/known colon cancer variants is needed. Assessing these in the context of sequencing depth requirements, sensitivity, and false positive rates is recommended.

A more detailed description of variants that make up the somatic truth set is needed. What is the distribution of variants across the genome? Does the 27Mb fall in coding regions? Etc.

What was the rationale for using Mutect2 for assessment of the truth set? (especially given the higher false positive rate) An ensemble approach may be useful here.

Minor comments:

Given the use of the LinSeq data set, it would be worth describing how much variation is observed across generations and the extent to which the truth set changes across cell divisions.

Reviewer #1 (Remarks to the Author):

Shand et al present a nice resource for somatic mutation variant calling (does the resource have a name? This might be useful). The goal is to overcome many of the limitations of previous synthetic resources for benchmarking somatic mutation calling, done by either mixing germline sequencing data or computationally spiking-in mutations. The main dataset is a known lineage of HT115 cells from the original LinSeq paper and their associated whole-genome sequencing data. The authors turn this into a resource for somatic mutation calling by mixing sequencing from cells (leaves of the tree) together to more accurately simulate heterogeneous tumor data. One challenging aspect of the paper is that it's hard to read without a lot of context from the original LinSeq paper. I have a couple of suggestions below to help make this more of a stand-alone effort, though there are probably many ways to do that. I also couldn't get the Terra link to load, so I'm not sure what sequencing datasets are available, but it would be nice if there were a full complement of calls for each leaf, all the mixtures, etc. to really highlight the resource aspect of this work. Overall this is a valuable resource for the community, but the paper needs significantly more details for end-users to find it useful.

We have clarified that the Terra link requires a free account and tested the link which currently works. We have added details of what is available at that link in the Data Availability section (line 307). We have also named the truth set LinST (Lineage derived Somatic Truthset).

Comments:

- How uniform are the clonal subpopulations? Given the cell line of choice here is a hypermutator, it would be nice to see summary stats on each leaf, mutational signatures (I would suggest anything but a 3d lego plot), and chromosome distributions. This is meant as a mutational calling resource, beyond the original LinSeq paper, and these are important details. Either a plot or an expansion of table 1. This would really help interpret figure 2.

Mutational signatures of Good Branch Variants for each leaf have been added in figure 4. Chromosome distributions and summary stats for the entire set of positive sites has been added in figure 3. Table 1 has also been expanded to include more details of the positive sites across each leaf.

- It's a bit of a can of worms, but could you expand on the "The allele fraction from Ludwig et al was not able to rediscover the known lineage tree on this dataset.." comment on line 245? It would be nice to understand how their approach failed here. Also if you're mentioning lineage it would probably be worth putting a review citation in for the emerging set of experimental lineage approaches.

We've expanded the section on using mitochondrial variant calls to determine the lineage on line 290. We've included a review citation for lineage approaches on line 286.

- The crux of the paper is the paragraph on line 93, which really needs to be expanded: For instance line (95) "For each depth and purity, this results in consistent sensitives and false positives per megabase, see Fig. 2." is all that's mentioned. How does the FP rate compare to TP? Or the number of calls overall? Also I don't think 'sensitives' is the right word (or a word)?

We've expanded figure 2 to include raw counts of false positives and true positives. We've also expanded and clarified that section on line 113.

- Can you compare Mutect2 calling statistics (FP,TP etc) to another synthetic calling resource (maybe the dream challenge)? It would be good to demonstrate that this callset overcomes many of the issues in previous datasets, even if it's only in one aspect.

We've added a comparison with DREAM looking at mutational signatures and variant classifications. This is included in figure 3 and addressed on line 122.

Reviewer #2 (Remarks to the Author):

In this study, Shand et al. generated a somatic truth set of somatic alterations from previously generated data from a colon cancer cell line. The development of strong somatic truth sets for analysis of WGS data continues to be an area of need. The use of more comprehensive analyses will be necessary and thus involves the use of cross-platform data to address and minimize deficiencies associated with sequencing platforms. However, this study does not address this aspect. The use of LinSeq data is interesting but is not clear how useful this data set would be for the research community, unless the data is meant to be a reference that can be used specifically for basic research studies utilizing the HT115 cell line.

It is not feasible for us to resequence these samples to obtain cross-platform data, but we have addressed this as potential future work on line 144. We've clarified the use of this data set beyond HT115 on line 42.

Major comments:

The use of tumor/normal pairs is a bit unclear and perhaps a revision to Fig1 would be helpful. Given that the goal of the study is to produce a somatic truth set, a major concern is that a true paired normal to HT115 was not sequenced/analyzed. Please clarify. Given cross-

generational genetic variation, capturing these changes in tumor and normal cells will be important for defining a somatic truth set. Along these lines, the reasoning for using distant sister tumor samples as “normals” is unclear.

We have added a panel to Figure 1 to clarify how the truth is constructed for an example pairing. Because we are using data from a previous paper, it is not feasible for us to sequence a matched normal or another HT115 sample that could act as a matched normal. This would definitely be helpful and is addressed on line 72 and line 134.

If the truth set is aimed to be used for colon cancer studies, comparison of the data against reported/known colon cancer variants is needed. Assessing these in the context of sequencing depth requirements, sensitivity, and false positive rates is recommended.

The truth set is aimed to be used more generally than colon cancer studies. We have included mutational signatures in Figure 3 and Figure 4 that are consistent with POLE mutant colon cancer samples to help describe the available true positives in the dataset. We hope this dataset can be useful to further understand the sequencing depth requirements (especially for different pipelines and tools), but it is beyond the scope of this paper to assess the current available pipelines.

A more detailed description of variants that make up the somatic truth set is needed. What is the distribution of variants across the genome? Does the 27Mb fall in coding regions? Etc.

We have expanded Table 1 to include details on where the variants are found (for example, how many sites exist in coding regions). We have also added the distribution of variant classifications across each chromosome in Figure 3. Mutational signatures are also now included in Figure 3 and Figure 4.

What was the rationale for using Mutect2 for assessment of the truth set? (especially given the higher false positive rate) An ensemble approach may be useful here.

The purpose of running a somatic mutation calling pipeline is to demonstrate that this truth set produces a reasonable estimation of sensitivity and precision. The goal is not to assess the tool itself. For that reason we have included another pipeline to broaden the results (VarScan2), but we don't find the need for an ensemble approach, because we are not trying to come up with the best somatic variant calling pipeline possible, we just want to assess this dataset as a truth set.

Minor comments:

Given the use of the LinSeq data set, it would be worth describing how much variation is

observed across generations and the extent to which the truth set changes across cell divisions.

The differences across cell divisions has been described in the original LinSeq paper. Since we are using all variants across generations in the final truth set it doesn't seem relevant to describe this variation here.

Thank you for your time and comments on this manuscript.

Sincerely,
Megan Shand
Computational Biologist
Data Sciences Platform
Broad Institute

On behalf of all authors.

Reviewers' comments:

Reviewer #1 (Remarks to the Author):

Again, Shand et al present a nice resource for somatic mutation variant calling, now called LinST. The goal is to overcome many of the limitations of previous synthetic resources for benchmarking somatic mutation calling, done by either mixing germline sequencing data or computationally spiking-in mutations. The main dataset is a known lineage of HT115 cells from the original LinSeq paper and their associated whole-genome sequencing data. The authors turn this into a resource for somatic mutation calling by mixing sequencing from cells (leaves of the tree) together to more accurately simulate heterogeneous tumor data. Overall the revisions to the paper are appropriate given the reviewer comments, it appears much improved. All the best,

Aaron

Reviewer #2 (Remarks to the Author):

The authors have addressed my original concerns. With the additional edits, I did have a few more questions & comments:

1) What's the rationale for using Strelka (& HaplotypeCaller) for identifying Good Branch & leaf variants, and Mutect2 & VarScan for the variant comparisons?

2) Paragraph starting @ line 111: The authors indicate that the high false positive to true positive rate is driven by Bad Branch Variants. If these variants are removed, what is the false positive to true positive rate?

3) It would be worth highlighting the mutational features of the LinST data, in particular the high level of TCT>TAT & TTT>TGT variants. For example, the former signature aligns with what is observed in the COSMIC SBS10a mutational signature, which is representative of POLE mutations. However, the latter signature is not as well defined such that discussion would be helpful. Describing these features helps to outline the context in which using the LinST data could be particularly useful for researchers.

Thank you for your comments on the manuscript. We have edited the manuscript to address your concerns. Please find a full list of point-by-point comments and responses below. All page numbers refer to the revised manuscript file with tracked changes.

1) What's the rationale for using Strelka (& HaplotypeCaller) for identifying Good Branch & leaf variants, and Mutect2 & VarScan for the variant comparisons?

We have added a comment about our reasoning for using these tools on line 203.

2) Paragraph starting @ line 111: The authors indicate that the high false positive to true positive rate is driven by Bad Branch Variants. If these variants are removed, what is the false positive to true positive rate?

This is an excellent point. We've added the average difference in false positives when removing the Bad Branch Variants from the truth and reworded this section to reflect those results. These changes are on line 116.

3) It would be worth highlighting the mutational features of the LinST data, in particular the high level of TCT>TAT & TTT>TGT variants. For example, the former signature aligns with what is observed in the COSMIC SBS10a mutational signature, which is representative of POLE mutations. However, the latter signature is not as well defined such that discussion would be helpful. Describing these features helps to outline the context in which using the LinST data could be particularly useful for researchers.

We have added the fact that the LinST data aligns with both COSMIC SBS10a and SBS28 on line 126.

Thank you for your time and comments on this manuscript.